

# Upper Stratospheric Temperature Trends: New Results from OSIRIS

Kimberlee Dubé[1], Susann Tegtmeier[1], Adam Bourassa[1], Daniel Zawada[1], Doug Degenstein[1],
William Randel[2], Sean Davis[3], Michael Schwartz[4], Nathaniel Livesey[4], and Anne Smith[2]

[1]Institute of Space and Atmospheric Studies, University of Saskatchewan, Saskatoon, SK, Canada
[3]NOAA Chemical Sciences Laboratory, Boulder, CO, USA
[2]National Center for Atmospheric Research, Boulder, CO, USA
[4]Jet Propulsion Laboratory, California Institute of Technology, Pasadena, California, USA

**Correspondence:** Kimberlee Dubé (kimberlee.dube@usask.ca)

**Abstract.** Temperature trends in the upper stratosphere, particularly above ∼45 km are difficult to quantify due to a deficit of long-term observational data in this region. The recent v7.3 upper stratospheric (35–60 km) temperature data product from the Optical Spectrograph and InfraRed Imager System (OSIRIS) includes over 22 years of observations that can be used to estimate temperature trends. The trends in OSIRIS temperatures over 2005–2021 are compared to those from two other satellite limb instruments: SABER and MLS. We find that the upper stratosphere cooled by ∼0.5 to 1 K/decade during this period. Results from the three instruments are generally in agreement. By merging the OSIRIS observations with those from channel 3 of the Stratospheric Sounding Unit (SSU), we find that the stratosphere cooled at a rate of approximately −0.6 K/decade between 1979 and 2021 near 45 km, in agreement with earlier results based on SSU and MLS. The similarity between OSIRIS temperature trends and those from other records improves confidence in observed upper stratospheric temperature changes over the last several decades.

## 1 Introduction

A consequence of increasing anthropogenic greenhouse gas emissions is an altered thermal structure in the atmosphere, consisting of tropospheric warming and stratospheric cooling (e.g., Manabe and Wetherald, 1967; Gulev et al., 2021). Temperatures in the troposphere and lower stratosphere (below ∼35 km) have been monitored for several decades by radiosondes (Haimberger et al., 2012) and satellites (Khaykin et al., 2017; Mears and Wentz, 2017), and temperature changes in this region are well defined (Ladstädter et al., 2023; Gulev et al., 2021). At higher altitudes, above ∼35 km, temperature observations are more limited, so there is uncertainty in the magnitude of the middle and upper stratospheric cooling rate (Gulev et al., 2021). New and updated temperature observations in the middle and upper stratosphere are necessary for better understanding the multidecadal cooling rate (cooling trend), and to more accurately quantify the impact of humanity on the climate. Considering middle and upper stratospheric cooling, rather than just tropospheric warming, increases the confidence that observed atmospheric temperatures are a direct result of human activities, and not due to natural variability (Santer et al., 2023).





Most knowledge about temperatures above ∼35 km comes from a series of nadir sounders that have operated on various National Oceanic and Atmospheric Administration (NOAA) satellites since late 1978 (Reale et al., 2008; Randel et al., 2009). Measurements are taken by three different instruments: the Stratospheric Sounding Unit (SSU), the Microwave Sounding Unit (MSU), and the Advanced Microwave Sounding Unit (AMSU-A). These instruments all have limited vertical resolution as temperatures are measured in different channels covering altitude ranges determined by their weighting functions (Randel et al., 2009). Channels 2 and 3 of SSU, and channels 13 and 14 of AMSU-A cover the range between ∼35 and ∼45 km, while MSU only has tropospheric and lower stratospheric channels. Each individual SSU and AMSU-A data record is quite short, and it is necessary to merge measurements from multiple instruments before calculating multidecadal trends (Zou et al., 2014). It is also necessary to merge the SSU observations with those from AMSU-A (or another instrument) when considering temperature trends over the full four decades from 1979 to the present: the last SSU instrument ceased operations in 2006, and the first AMSU-A instrument began operating in 1998 (Zou and Qian, 2016).

Satellite limb instruments are the best option available for retrieving temperature profiles that have a high (1–4 km) vertical resolution and extend into the upper stratosphere. Limb observations have been available since the end of the 20th century from an assortment of instruments. Datasets from a single instrument that extend for multiple decades, such as the Atmospheric Chemistry Experiment - Fourier Transform Spectrometer (ACE-FTS, 2004/2–, Bernath et al., 2005; Boone et al., 2020), the Microwave Limb Sounder (MLS, 2004/8–, Waters et al., 2006; Schwartz et al., 2008), and the Sounding of the Atmosphere Using Broadband Emission Radiometry instrument (SABER, 2002/1–, Russell et al., 1999; Remsberg et al., 2008), are best when considering atmospheric trends. Randel et al. (2016) also created a merged SSU+MLS data record covering 1979 to the present.

Global mean temperature trends in merged SSU+AMSU+A and SSU+MLS datasets for the ozone recovery period (post ∼1998) range from -0.19 K/decade to -0.5 K/decade (SSU channel 2) and from -0.28 K/decade to -0.6 K/decade (SSU channel 3) (Randel et al., 2016, 2017; Maycock et al., 2018; Steiner et al., 2020). The disparate time periods and latitude regions that were used make it difficult to compare the cooling rates from different studies directly, but in general the cooling rate is greater at higher altitudes and including more recent years in the analysis (e.g., Steiner et al., 2020) results in a greater stratospheric temperature decrease per decade compared to older studies (e.g., Randel et al., 2017).

Here we focus on results from a new temperature retrieval in the middle and upper stratosphere (35–60 km) that was recently developed for the Optical Spectrograph and InfraRed Limb Imager (OSIRIS, Llewellyn et al., 2004; Zawada et al., 2024). OSIRIS has been in orbit on Odin since 2001, and the 22+ year data record provides an excellent opportunity to study long-term cooling in the middle and upper stratosphere. In the first part of this work, OSIRIS temperature trends are compared to those from SABER and MLS. The observation-based temperature trends are also compared to temperature trends from several reanalyses and a climate model in order to assess the ability of models and data assimilation products to represent upper stratospheric cooling. The second main goal of this work is to create a merged SSU+OSIRIS temperature product, to complement the existing SSU+MLS and SSU+AMSU-A datasets (Randel et al., 2016; Zou and Qian, 2016). By merging more recent observations with those from SSU, which operated from 1979 to 2006, it is possible to look at changes in stratospheric





temperatures over more than four decades. Considering each of the OSIRIS, MLS, and AMSU-A observations for the last 20 years of the record provides increased confidence in observed temperature trends during the 21st century.

## 2 Data and Models

### 2.1 Satellite Observations

#### 2.1.1 OSIRIS

The optical spectrograph component of OSIRIS measures limb-scattered sunlight between 280 nm and 810 nm, with a spectral resolution of approximately 1 nm. Each scan takes about 90 seconds, and there are 15 orbits per day, resulting in 100–400 vertical solar irradiance profile measurements each day, depending on the time of year and the scanning mode. Temperature profiles are retrieved from the limb scatter measurements by first calculating the Rayleigh scattering number density at 310 nm

and 350 nm and then using the hydrostatic balance and the ideal gas law to convert the number density to temperature. The temperature calculations require ancillary temperature data at 65 km to initialize the profiles. Specific details about the OSIRIS v7.3 temperature retrieval are given in Zawada et al. (2024).

There are two versions of the OSIRIS temperature product: one that uses a value from the Modern-Era Retrospective analysis for Research and Applications, Version 2 (MERRA-2, Gelaro et al., 2017) interpolated to the OSIRIS profile as a reference

temperature at 65 km, and one that uses climatological values from the NRLMSISE-00 model (Picone et al., 2002) as the reference temperature. The choice of reference temperature is the main source of uncertainty in the OSIRIS retrieval above 45 km: it introduces a bias of up to 5 K at 65 km that decreases exponentially with decreasing altitude (Zawada et al., 2024). The MERRA-2 version of the retrieval is more physically realistic as the climatology forces a trend of 0 K/decade at 65 km, so the MERRA-2 based OSIRIS retrieval is used as the default. The effect of the reference temperature choice on the OSIRIS

temperature trends is discussed further in Section 4.

Only the OSIRIS descending node profiles are used due to a drift in Odin's orbit that has resulted in a loss of ascending node measurements over the course of the mission. The descending node observations occur near a local solar time (LST) of 6:30 am. The data are further filtered by removing scans with a solar zenith angle greater than $85°$. Monthly zonal means are then calculated for months with more than 15 measurements in a given 10 degree latitude and 1 km altitude bin. Months with fewer

profiles typically occur when OSIRIS resumes taking measurements after being in darkness (i.e. following the winter at mid and high latitudes).

#### 2.1.2 MLS

MLS has been operating from the Aura satellite since August 2004 (Waters et al., 2006). MLS observes microwave limb emissions, measuring $\sim$3500 vertical profiles each day. Temperatures are retrieved near the $O_2$ spectral lines at 118 GHz and

85 239 GHz (Livesey et al., 2022). The vertical resolution of the temperature profiles is 3 km at 30 hPa ($\sim$25 km), and decreases to 9 km at 0.1 hPa ($\sim$65 km) (Schwartz et al., 2008). Temperatures from version 5 of the MLS retrieval are used here. All profiles





are filtered per the guidelines provided in Livesey et al. (2022). As MLS is retrieved on a native pressure grid, the profiles must be converted to a vertical altitude grid before comparison with OSIRIS. This is done using the geopotential height (GPH) profiles that are retrieved along with each MLS temperature profile to calculate the geometric height of each pressure level,

and then interpolating to the 1 km OSIRIS altitude grid.

### 2.1.3 SABER

SABER measures infrared $CO_2$ emissions from its platform aboard the Thermosphere, Ionosphere, Mesosphere Energetics and Dynamics (TIMED) satellite. It has been in orbit since December 2001. Temperatures are retrieved between 10 km and 100 km, with a vertical resolution of 2 km (Remsberg et al., 2008). Profiles from version 2.0 of the SABER retrieval are used

here. The SABER viewing mode is such that continuous coverage is available from 52°S to 52°N, with higher latitudes observed for 60–63 day periods that alternate between the hemispheres (Remsberg et al., 2008). To avoid any bias that this might introduce in the trends we only use SABER observations from 50°S–50°N.

Due to the sampling pattern, the SABER measurement time changes with each scan, rather than observing near a fixed LST like OSIRIS and MLS. It takes approximately 60 days for SABER to observe the full 24-hour cycle. This could introduce a

100 bias when considering monthly mean temperatures as only half of the LSTs will be sampled. We tried accounting for this by using 30 days on either side of the 15th of each month to calculate the monthly zonal means, such that the full range of LSTs was included in the mean, as suggested by Zhao et al. (2021). It was found that the temperature trends were nearly the same whether regular monthly means (averaging from the first to last day of a month) were used, or this more complicated technique, so only results for the regular monthly means are shown in Section 4.

### 2.1.4 SSU and AMSU-A

The SSU and AMSU-A instruments were designed to be used for weather forecasting, but the global coverage and extensive length of the data record allows their observations to be used for climate-length trend studies. SSU is a three-channel radiometer that measures infrared $CO_2$ emissions. The weighting functions of the channels peak near 28, 36, and 45 km, respectively (Miller et al., 1980). SSU instruments were flown on numerous NOAA satellites between November 1978 and April 2006. We

use the NOAA Version 2 SSU temperature dataset developed by Zou et al. (2014). This version of the data uses reprocessed temperatures retrieved from recalibrated radiances, which improved agreement between SSU observations taken from different spacecraft.

AMSU-A measures molecular oxygen emissions between 50 and 58 GHz (Zou and Qian, 2016). AMSU-A has higher vertical resolution than SSU. There are 15 channels, with channels 9-14 dedicated to measuring temperatures at approximately

115 18, 20, 25, 30, 35 and 40 km. Various iterations of AMSU-A have flown on NOAA, NASA, and MetOp spacecrafts since 1998. The process for combining these observations into a single record is described in Wang and Zou (2014).

We also consider two merged stratospheric temperature datasets that use the SSU measurements. The SSU+AMSU-A dataset created by Zou and Qian (2016) uses a merging process that combines information from multiple AMSU-A channels to weight the higher resolution AMSU-A observations such that they match the three SSU channels. Randel et al. (2016) combined the



SSU temperature observations with temperature retrieved from MLS. The much higher vertical resolution of MLS compared
to SSU means that the MLS profiles can simply be weighted with the SSU weighting functions, before using the overlap period
to combine the datasets. Randel et al. (2016) and Steiner et al. (2020) found that trends in the SSU+MLS record agreed with
trends in SSU+AMSU-A temperatures within the regression uncertainties.

## 2.2   Reanalyses and Climate Model

The observed temperature trends are compared to reanalysis and model results to evaluate the ability of these systems to
accurately represent changes in upper stratospheric temperatures. The lack of temperature observations above 45 km (prior to
∼2004) makes it particularly difficult to evaluate model simulations in this region.

    The three most up-to-date reanalyses are considered: MERRA-2, ERA5, and the Japanese 55-year Reanalysis (JRA-55).
MERRA-2 is the latest reanalysis from the NASA Global Modelling and Assimilation Office (GMAO), based on the God-
dard Earth Observing System (GEOS) Model (Gelaro et al., 2017). JRA-55 is produced by the Japan Meteorological Agency
(JMA) (Kobayashi et al., 2015). ERA5 is the fifth generation reanalysis from the European Centre for Medium Range Weather
Forecasting (ECMWF) (Hersbach et al., 2020). ERA5, JRA-55, and MERRA-2 all assimilate radiances from SSU, MSU, and
AMSU, as well as bending angles from GNSS-RO instruments (Gelaro et al., 2017; Hersbach et al., 2020; Kobayashi et al.,
2015). MERRA-2 also assimilates MLS temperatures, which are included above 5 hPa beginning in August 2004 (Gelaro et al.,
2017). It should be noted that despite including many of the same observations, each reanalysis deals with the transitions be-
tween satellites and instruments in a different way. These transitions, along with changes in the reanalysis production streams,
can create discontinuities that occur at different times in each reanalysis (Long et al., 2017; Fujiwara et al., 2017).

    A cold bias in the stratosphere exists in ERA5 between 2000 and 2006 (Simmons et al., 2020). This motivated the de-
velopment of a corrected reanalysis for those years, called ERA5.1. For simplicity, when we refer to ERA5 we are actually
referring to the combined ERA5/ERA5.1 dataset. It should be noted that while ERA5.1 is generally an improvement, Simmons
et al. (2020) found that the combination of ERA5 and ERA5.1 does not perform as well as the previous generation reanalysis,
ERA-Interim, with regards to upper stratospheric temperatures for years prior to ∼2010.

    The OSIRIS temperatures are retrieved on an altitude grid with 1 km spacing, so the reanalysis results must be converted to
this same grid before doing any comparisons. First, reanalysis temperatures are interpolated to the latitude, longitude, and time
of each OSIRIS profile. In the case of MERRA-2 we start with the 3-hourly temperature profiles on pressure levels and use the
corresponding geopotential height to compute the geometric altitude corresponding to each pressure level for each profile. This
relationship is then used to interpolate the temperature profiles to the OSIRIS altitude grid. For ERA5 we start with the hourly
model level results and calculate the geopotential height of each model level from the surface pressure, before computing the
geometric altitude of each level and interpolating to the OSIRIS grid. For JRA-55 we use 6-hourly model level results. The
geopotential height on each model level is provided, so we only have to calculate the geometric altitude and interpolate to the
OSIRIS profiles locations/times. The same process is repeated for ERA5, JRA-55, and MERRA-2 but interpolated to the MLS
profile locations and times so that we can determine the impact of the OSIRIS sampling on the resulting trends.





In addition to the three reanalyses, we also consider temperature trends from simulations using the Whole Atmosphere Community Climate Model (WACCM) version 6 (Gettelman et al., 2019). WACCM6 has 70 vertical levels extending from the surface to 140 km and a horizontal resolution of 0.95° latitude by 1.25° longitude. We consider four ensemble members from the free-running version of the model, covering the period 1960–2018, and following the REFD1 scenario. This scenario includes forcing from observed sea surface temperatures, greenhouse gases, ozone depleting substances, and volcanic aerosol (Plummer et al., 2021). The Quasi-Biennial Oscillation (QBO, Wallace et al., 1993) was nudged to match observations.

## 3 Regression Analysis

A multiple linear regression (MLR) model is applied to monthly zonal mean observations in 10 degree latitude and 1 km altitude bins to study the long-term trends and variability in upper stratospheric temperatures. The MLR model is defined as

$$T(t) = \beta + \beta_{trend} \times linear(t) + \beta_{qboa}^{(2)} \times QBO_a(t) + \beta_{qbob}^{(2)} \times QBO_b(t) + \beta_{solar} \times F10.7(t) + R(t), \tag{1}$$

where each $\beta_i$ defines a regression coefficient. The superscripts in Equation 1 define the highest order seasonal harmonic included for a term. Harmonics are included for the QBO predictors, $QBO_a(t)$ and $QBO_b(t)$, in order to account for coupling between the QBO and the seasonal cycle. Thus, the coefficient for the $QBO_a(t)$ term is

$$\beta_{qboA}^{(2)} = \beta_{qboA}^0 + \sum_{k=1}^{2} \left( \beta_{qboA}^{2k-1} \sin \frac{2\pi}{365.25} kt + \beta_{qboA}^{2k} \cos \frac{2\pi}{365.25} kt \right). \tag{2}$$

$\beta_{qboB}^{(2)}$ is also expanded in the same way. There are 13 regression coefficients in total: three corresponding to the constant, trend, and $F10.7(t)$ terms in Equation 1, plus five coefficients from each of the $QBO_a(t)$ and $QBO_b(t)$ terms. The data are deseasonalized prior to applying the regression model, so there is no need to include regression terms for annual oscillations. The deseasonalization is done to monthly zonal mean data by subtracting the mean temperature of a given month from all values for that month for a specified latitude and altitude bin.

In Equation 1, $\beta_{trend}$ is the temperature trend in units of K/decade, and $F10.7(t)$ is the solar flux at 10.7 cm. The MLR was also tested with terms representing the El-Niño Southern Oscillation and the aerosol optical depth, however these were found to play a negligible role in explaining the temperature variability between 35 and 60 km. Further details on the regression model, as well as the proxy data sources, are described in Damadeo et al. (2022).

We only consider temperatures to the end of 2021 when calculating trends to avoid the influence of the Hunga Tonga-Hunga Ha'apai (HTHH) Volcanic Eruption, which significantly altered stratospheric temperatures throughout 2022 (Wang et al., 2023; Yu et al., 2023). As this occurred near the end of our dataset it could skew the trend values by altering the end point. Including an aerosol optical depth proxy in the MLR is not adequate to account for the effects of HTHH as water vapour played a significant role in altering the dynamics and composition of the stratosphere following the eruption.

OSIRIS measures limb-scattered sunlight, so there are only observations available during daylight. This means that there are no data available when the measurement time of the descending node (local time of approximately 6:30 AM) occurs during the night, i.e. at higher latitudes in the winter. In more recent years there are also some gaps in the monthly mean observations





because the aging OSIRIS instrument does not have power for as much of each orbit as it used to. These months without
OSIRIS measurements, which are different for each latitude bin, are removed from the MLS and SABER observations before
applying the MLR, in order to most directly compare trends in all three datasets. The effect on the trends of removing these
points, as well as of the overall OSIRIS sampling pattern, is discussed in Section 4.

## 4   Results

### 4.1   Vertically resolved temperature trends

An initial validation of the OSIRIS temperature observations is provided in Zawada et al. (2024). MLS and OSIRIS tem-
peratures were shown to agree within $\pm5$K between 35 km and 55 km, with some of the bias caused by differences in the
measurement time of day. The OSIRIS and MLS time series also agree very well: Figure 1 shows the deseasonalized tempera-
ture anomalies for OSIRIS, MLS, and SABER in four example latitude/altitude bins. The variability is similar across all three
datasets, and the correlations are greater than 0.5 in all bins, and greater than 0.8 in most bins below 45 km (Appendix, Figure
A1). MLS and SABER are more similar to one another than either is to OSIRIS: much of this difference is likely due to the
sparser OSIRIS sampling pattern.

In the tropics the largest source of variability up to $\sim$45 km is the QBO (Figure 1, panel B). At latitudes greater than $\pm40°$
the magnitude of the temperature anomalies peaks in the winter, and lasts to the spring. Only the tail end of these peaks, in
September, are visible in Figure 1 panel C due to the lack of OSIRIS observations in the winter at 50°S. At this time of year
there is significant interannual variability in the temperatures that does not get removed when deseasonalizing the data.

The MLR described in Section 3 is used to determine temperature trends over 2005–2021 (2005 is the first full year when all
three instruments were operating). The trends are shown in Figure 2. Observations from each of MLS, SABER, and OSIRIS
show stratospheric cooling during 2005–2021, ranging from about -0.5 K/decade to -1.5 K/decade. OSIRIS observations have
the greatest cooling in the southern hemisphere (SH) and tropics, while SABER observations show the greatest cooling in the
northern hemisphere (NH). Despite this difference, the OSIRIS and SABER temperature trend profiles (Figure 2, panels D to
H) have a similar vertical structure, particularly in the tropics (panel F). When considering the larger latitude bins, OSIRIS
and SABER temperature trends agree within the regression uncertainty everywhere except at 50 km and above 56 km in the
30°S-10°S bin.

The MLS temperature trends agree with those from OSIRIS and SABER in the tropics, but at higher latitudes the MLS
trends oscillate in altitude (Figure 2, panel D and H): at the stratopause (48-50 km) the MLS cooling rate is $\sim$-1.5 K/decade,
but the trend quickly drops to nearly zero K/decade at 45 km, before going back to $\sim$-1 K/decade at 40 km. The effect is more
pronounced in the SH compared to the NH. The OSIRIS and SABER trends change very little between 40 and 50 km in either
hemisphere. More work is required to determine if the vertical structure in the MLS trends is physical.

The MLR used to compute the temperature trends also includes terms for the QBO and the solar cycle. The regression coef-
ficients corresponding to these terms for OSIRIS, SABER, and MLS are provided in the Appendix, Figure A2. The coefficients
are very similar for all three datasets. The solar cycle, represented by the F10.7 solar flux proxy, has a positive impact on



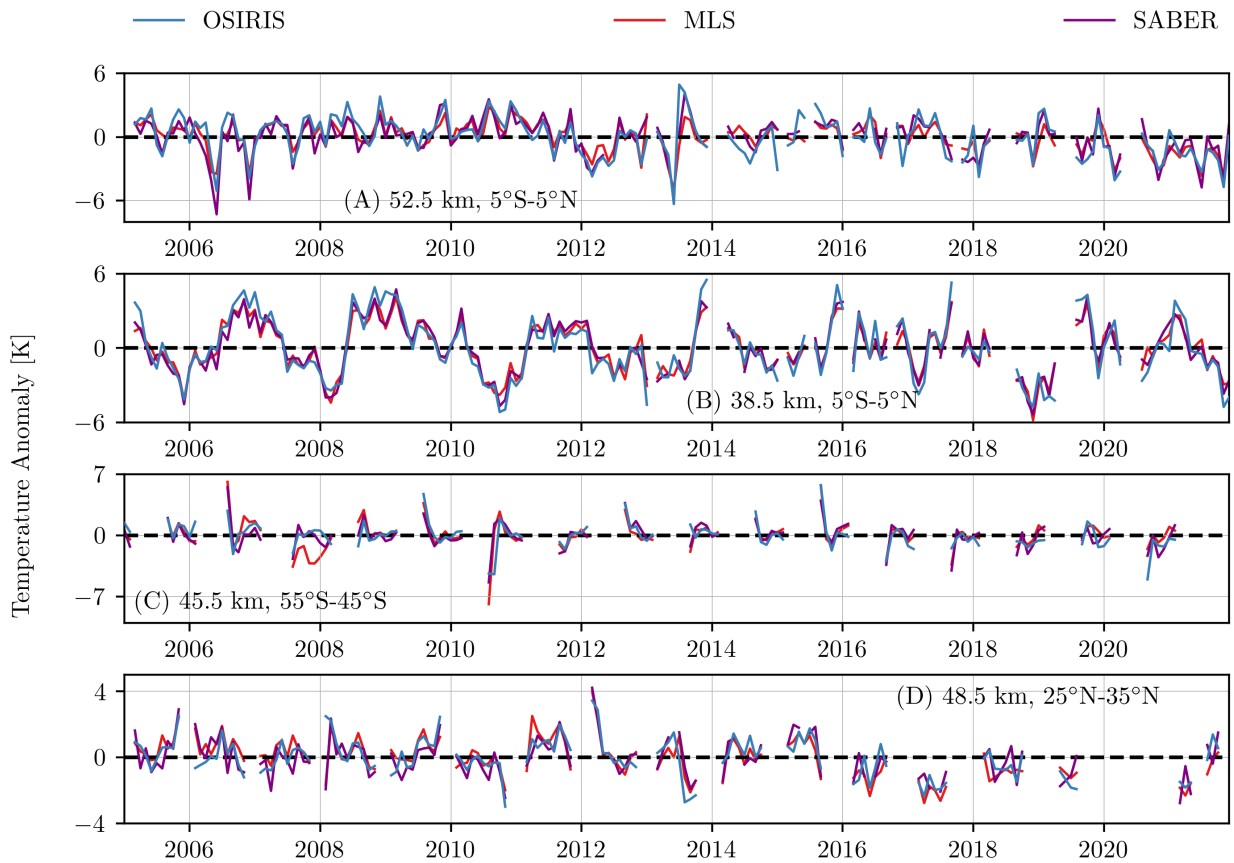

**Figure 1.** Deseasonalized monthly zonal mean anomaly of OSIRIS, SABER, and MLS temperatures in four representative 10 degree latitude and 1 km altitude bins. Results are shown only for months when data from all three instruments are available.

the temperature throughout the upper stratosphere as expected, as higher levels of solar irradiance lead to greater warming. The high values for the QBO coefficients in the SH are caused by the OSIRIS sampling pattern, and not by a real physical phenomena. OSIRIS only measures at higher latitudes in the SH for a few months of the year. When months without OSIRIS observations are not removed from MLS and SABER, the SH QBO coefficients for these two data sets look more similar to their NH counterparts (not shown here)

There are two main factors that introduce uncertainties into the OSIRIS temperature trends: the choice of the reference temperature used in the retrieval, and the spatial/temporal sampling pattern. We quantify how the choice of reference temperature influences the trends by comparing the trends in OSIRIS temperatures retrieved using MERRA-2 to the trends in temperatures retrieved using NRLMSISE-00 reference temperatures. NRLMSISE-00 is a climatology, and there is no trend (0 K/decade trend) in the temperatures at the reference altitude of 65 km, even after interpolating to the OSIRIS profiles locations/times. Therefore the difference in the trends for the two OSIRIS retrieval versions shows how much MERRA-2 is contributing to



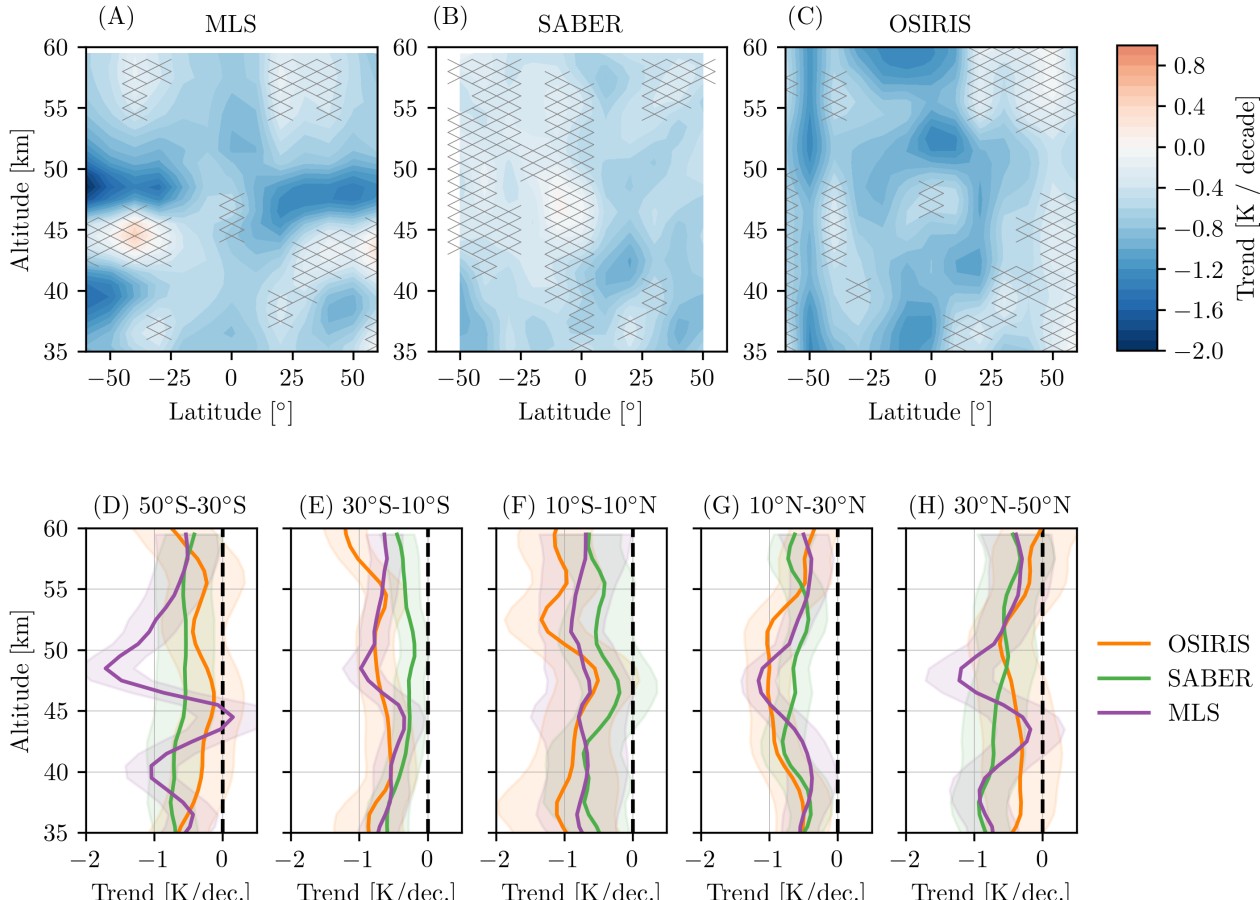

**Figure 2.** Temperature trends for 2005–2021. Trends are shown for (A) MLS, (B) SABER, and (C) OSIRIS. Hatching denotes statistically insignificant trends at the $2\sigma$ level. The bottom row, panels (D) to (H), shows vertical profiles comparing the same trends from the three instruments in 20 degree latitude bins. The shaded regions denote the $2\sigma$ uncertainty in the MLR.

the resulting OSIRIS temperature trend at each latitude/altitude (Figure 3). The influence of the reference temperature on the retrieved temperatures decreases exponentially downward in altitude, becoming small below ∼45 km (Zawada et al., 2024).

Similarly, the effect of the reference temperature on the temperature trends is greatest at 60 km, and negligible below ∼45 km. The reference temperature does not have the same impact on the OSIRIS trends at all latitudes. This is because the trend in MERRA-2 at the reference altitude is more negative in the SH compared to the NH, resulting in an OSIRIS trend that is further away from the climatological 0 K/decade trend in the SH. Overall, the effect of the reference temperature on the trends is less than 0.3 K/decade below 50 km in the NH/tropics and at almost all levels in the SH. It is important to note that the effect of

the reference temperature trend does not correspond directly to an error in the retrieved OSIRIS trends: MERRA-2 assimilates



MLS observations so the temperatures are more physically realistic than those from a climatology. Comparing the temperature trends from the two versions of the OSIRIS retrieval only tells us how much of an effect the reference temperature choice can have on the trends.

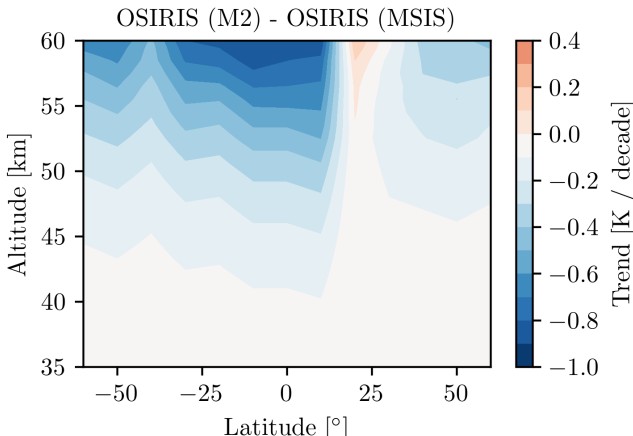

**Figure 3.** The difference between temperature trends from OSIRIS retrieved with a MERRA-2 reference temperature, and OSIRIS retrieved with a climatological (NRLMSISE-00) reference temperature. Trends are calculated over 2005–2021.

To evaluate the impact of the OSIRIS sampling pattern on the temperature trends we compare trends in reanalysis tempera-
tures that are sampled like OSIRIS and that are sampled like MLS. The OSIRIS temperature trends and the trends in each of MERRA-2, ERA5, and JRA-55 sampled to the OSIRIS profiles are shown in the top row of Figure 4. The middle row of the Figure shows the MLS temperature trends and trends in the same three reanalyses but sampled like MLS. The MLS trends are slightly different from those in Figure 2 as months when OSIRIS does not have any observations were removed from MLS before calculating the trends in Figure 2. The differences in the reanalysis trends with MLS sampling compared to OSIRIS
sampling are in the bottom row of Figure 4. This direct comparison shows that the effect of the OSIRIS sampling pattern on the trends is largest at latitudes greater than $\pm 30°$. The effect of sampling is also slightly greater in the SH compared to the NH. As OSIRIS can only measure the sunlit portion of the atmosphere, there are regularly gaps in the data record at mid-high latitudes, depending on the season, so it is logical for the sampling pattern to affect the trends more at these latitudes. The OSIRIS orbit is also such that there are more observations in the NH compared to the SH, resulting in the greater impact of
sampling on the SH trends. As with the reference temperature, it is not possible to relate these sampling biases directly to an error in the OSIRIS trends, we can only conclude that caution should be taken when considering latitudes greater than $\pm 30°$, particularly above 50 km.

In addition to their use for discussing the OSIRIS sampling issues, the reanalysis temperature trends are also worth consider-
ing on their own. Reanalyses are often taken to be the best measure of the truth when validating and tuning climate models, but
they are limited by the data that is assimilated and often have discontinuities whenever there are changes in the observational records that are assimilated (Long et al., 2017). MERRA-2 is the only reanalysis that assimilates temperatures above ~45





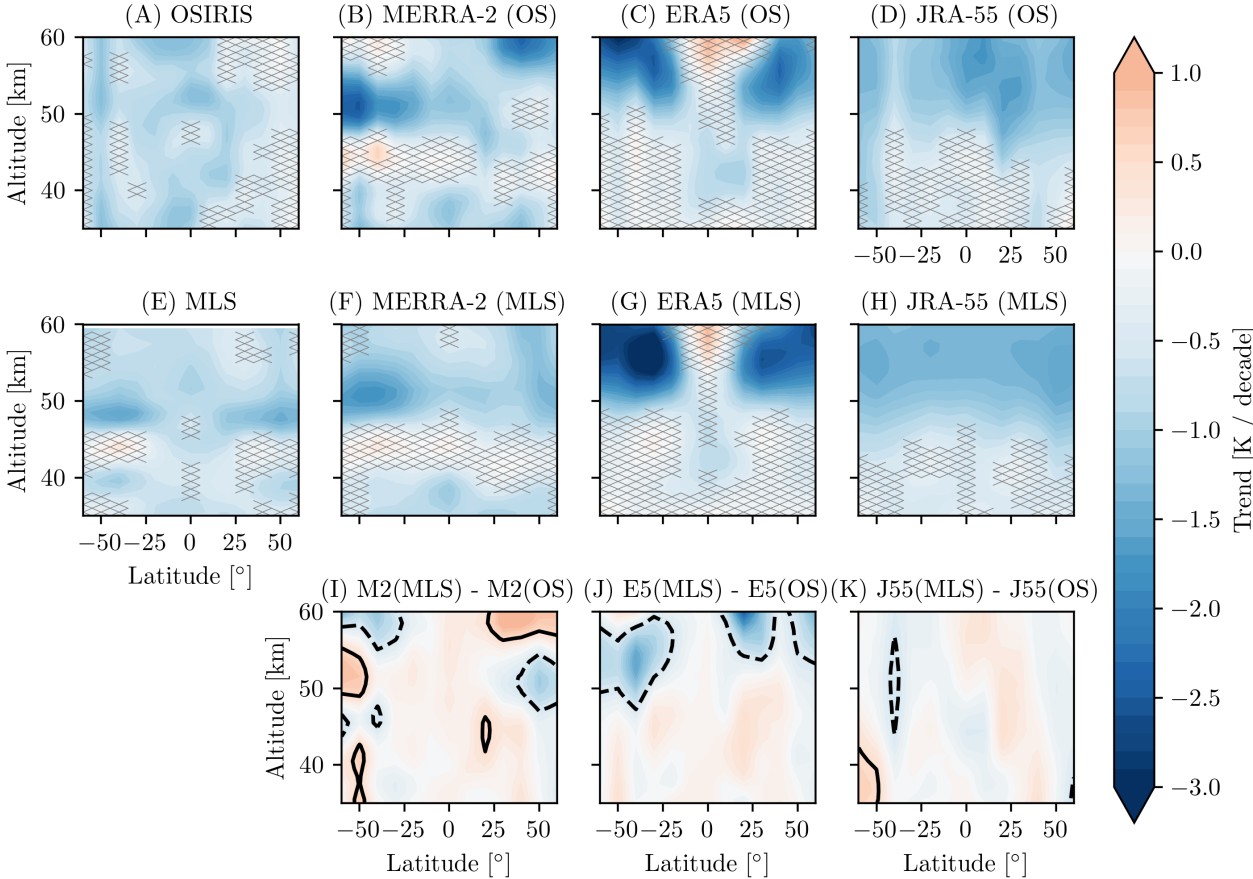

**Figure 4.** Temperature trends for 2005–2021. Trends are shown for (A) OSIRIS and (E) MLS, along with MERRA-2, ERA5, and JRA-55 sampled like OSIRIS (B, C, D) and sampled like MLS (F, G, H). Panels (I, J, K) show the difference between trends with the two types of sampling for each reanalysis. The black contour lines mark differences of -0.5K/decade (dashed) and +0.5 K/decade (solid). In all panels hatching denotes statistically insignificant trends at the $2\sigma$ level.

km, corresponding to the upper limit of SSU/AMSU-A observations. MLS temperatures are assimilated above ∼30 km, and the result is that the MERRA-2 temperature trends are very similar to the MLS temperature trends. The ERA5 temperature trends look similar to those in OSIRIS and MLS in the tropics below 45 km where there is data assimilated, but the cooling rate at higher altitudes is more than twice what is seen in observations. JRA-55 also does not assimilate MLS, but the JRA-55 temperature trends above 45 km are nonetheless more similar to MERRA-2 and the observations than they are to ERA5. This suggests that the problem with ERA5 is not solely because of the lack of assimilated observations at higher altitudes. There are discontinuities in the ERA5 temperature time series at the higher altitudes that contribute to the more negative trends. Further




work is needed to determine the origin of these discontinuities in ERA5 as they are not obviously related to changes in the
processing or in the assimilated observations.

Finally, we consider temperature trends in 4 ensemble members from the WACCM REFD1 scenario (Appendix, Figure A3). The WACCM results are only available to the end of 2018, so these trends cannot be compared directly with the results from observations and reanalyses. The key point here is rather that the temperature trends from each WACCM ensemble member have substantial variability, of up to 2 K/decade in some latitude/pressure bins. Since the emissions and radiative calculations
are identical in each ensemble member, this suggests that upper stratospheric temperature trends are significantly affected by internal variability (over the relatively short period of 2005–2021).

### 4.2  Merging with SSU and trends in SSU channel 3

It is necessary to combine observations from multiple instruments to study upper stratospheric temperature trends prior to the 21st century. The most consistent data source available to use is SSU, which operated from 1979 to 2006. As discussed in
Section 2, SSU temperature have been previously merged with those from MLS and AMSU-A. We now create a third merged dataset using OSIRIS temperatures.

Before they can be merged with SSU temperatures, it is necessary to weight the OSIRIS temperature profiles using the SSU weighting functions. We only consider SSU channel 3, as it is the channel that best matches the OSIRIS altitude range: 82% of the channel 3 weighting function falls between 35 km and 60 km (Figure 5). As another point of comparison, we also weight
the OSIRIS temperature profiles using the narrower AMSU-A channel 14 weighting function, 92% of which falls within the OSIRIS altitude range.

Each OSIRIS profile is individually weighted by the weighting functions before calculating the monthly zonal means. Panel A of Figure 6 shows the mean bias between temperatures from SSU channel 3 with OSIRIS, and between temperatures from AMSU-A channel 14 and OSIRIS. In both cases the bias is slightly higher in the tropics compared to other latitudes. OSIRIS
and SSU agree within 6-7 K, while OSIRIS and AMSU-A agree with 1-2 K. Panels B-D compare the OSIRIS and SSU time series at Northern and Southern mid-latitudes and in the tropics. While the monthly variability is similar, OSIRIS is consistently biased high. Panels E-G of Figure 6 show the same comparison for AMSU-A and OSIRIS. The datasets are extremely similar, and there are no changes in the bias with time, which provides confidence that the various AMSU-A datasets were merged correctly. It is likely that OSIRIS agrees better with AMSU-A than with SSU because AMSU-A has narrower weighting
functions, and AMSU-A channel 14 aligns better with the OSIRIS retrieval range than SSU Channel 3. While AMSU-A is not particularly useful for extending the OSIRIS observations as the measurement periods are nearly the same, the similarity between OSIRIS and AMSU-A provides further confidence in the accuracy of the OSIRIS temperature retrieval, at least below 45 km.

After weighting the OSIRIS profiles with the SSU channel 3 weighting function, the merging process is the same as the one
used to merge OSIRIS $O_3$ and $NO_2$ with observations from the Stratospheric Aerosol and Gas Experiment II (Bourassa et al., 2014; Dubé et al., 2020). First, the bias between the OSIRIS and SSU temperatures is removed by subtracting the bias from OSIRIS in each latitude bin. Then the datasets are deseasonalized individually, before merging by taking the mean in months

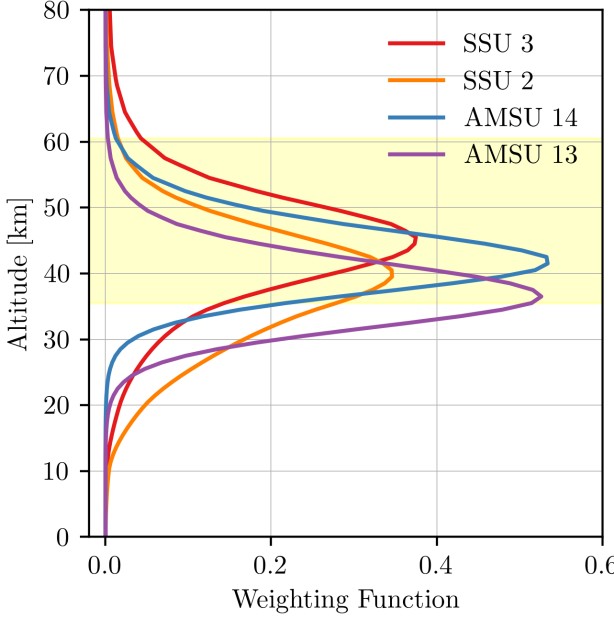

**Figure 5.** Weighting functions for SSU channels 2 and 3 and AMSU-A channels 13 and 14. The yellow shaded region denotes the altitude range of the OSIRIS temperature product.

when both instruments have observations. The resulting time series, for several 10 degree latitude bins, are shown in Figure 7. While the OSIRIS sampling affects the results somewhat, the SSU+OSIRIS temperatures are extremely similar to both the

SSU+MLS and SSU+AMSU temperatures at all latitudes.

Trends in each of the merged datasets are calculated using the MLR described in Section 3. Panel A of Figure 8 shows the temperature trends as a function of latitude during between 1979 and 2021. The trends from all three merged datasets are nearly identical. At this level, near 45 km, the stratosphere cooled by ∼0.6 K/decade during the 42 years considered. The cooling rate is slightly greater in the NH than in the SH. For just the OSIRIS period, from 2002–2021, the merged temperature records are

mainly based on the other instrument, rather than SSU, so we are comparing MLS, AMSU-A, and OSIRIS trends only. These temperature trends agree within the regression error at all latitudes (Panel B of Figure 8). In a few bins the OSIRIS temperature trends are less similar to MLS and AMSU-A, most likely because of sampling differences. At all latitudes the cooling rate is about 0.5 K/decade.

## 5   Conclusions

Upper stratospheric temperature trends have historically been difficult to quantify due to a deficit of observations above 35 km. Using the new OSIRIS v7.3 temperature product, we find that the upper stratosphere, between 35 and 60 km, cooled







**Figure 6.** Panel A: Bias between OSIRIS and SSU, and OSIRIS and AMSU-A temperature for the overlap periods. OSIRIS is weighted separately to match SSU Ch. 3 and AMSU-A Ch. 14. Panels B, C, D: SSU Ch. 3 temperatures and OSIRIS temperature weighted like SSU Ch. 3 for three latitude bands. Panels E, F, G: AMSU-A Ch. 14 temperatures and OSIRIS temperature weighted like AMSU-A Ch. 14 for three latitude bands.





**Figure 7.** Merged SSU+AMSU-A, SSU+MLS, and SSU+OSIRIS temperature anomalies for five latitude bands. All datasets are weighted like SSU Ch. 3.

by 0.5–1 K/decade during 2005–2021. The two main sources of uncertainty in the OSIRIS temperature trends are due to sampling biases and the choice of the reference temperatures used in the OSIRIS retrieval. These factors somewhat limit our confidence in the OSIRIS temperature trends at latitudes greater than $\pm30°$ and at altitudes above 50 km. Despite this, the
OSIRIS temperature trends agree with trends from SABER and MLS within the regression uncertainties at most latitudes and altitudes between $\pm50°$ and 35–60 km. By having a third temperature record in the upper stratosphere, where previously there were only MLS and SABER, we increase confidence in the stratospheric cooling rate. We are also able to observe a possible



**Figure 8.** Trends in temperatures from merged SSU+AMSU-A, SSU+MLS, and SSU+OSIRIS in ten degree latitude bins. Trends are shown for (A) 1979–2021 and (B) 2002–2021. Error bars are the $2\sigma$ uncertainty in the MLR.

issue with MLS temperatures at latitudes outside $\pm30°$. At these latitudes the MLS temperature trends oscillate in altitude, with trends becoming significantly more negative than those from SABER and OSIRIS near 50 km.

We also compared the OSIRIS and MLS temperature trends to temperature trends from reanalyses and a climate model. The modelled temperature trends from four WACCM ensemble members are generally within the range of those from the observational datasets, but internal variability alters the trends by up to 2 K/decade, highlighting trend uncertainties in short




data records. The reanalysis trends agree reasonably well with the observations below 45 km, where SSU and AMSU-A observations are assimilated, but are highly variable at higher altitudes. MERRA-2 is the only reanalysis that assimilates temperatures (from MLS) above 45 km, and it is clear that from the large trend differences with ERA5 and JRA-55 that this constraint is important. However, this is not the only factor affecting the reanalysis temperatures trends above 45 km. JRA-55 trends are at most ~1 K/decade too low, while ERA5 trends in some bins are more than 3 K/decade lower than the trends in observations. This suggests that there is some issue with the ERA5 temperatures at these altitudes, apart from the lack of assimilated observations. The ERA5 temperature time series has discontinuities above ~54 km that contribute to the more negative trends. Further work is needed to understand these discontinuities as they cannot be clearly attributed to changes in the production stream or to changes in the input observations.

For the comparison of OSIRIS temperature observations to those from the nadir sounders SSU and AMSU-A, the OSIRIS profiles were weighted to match either channel 3 of SSU or channel 14 of AMSU-A. OSIRIS and AMSU-A temperatures agree extremely well, with OSIRIS biased high by at most 2 K. The bias between OSIRIS and SSU channel 3 is greater: OSIRIS is warmer by 6–7 K.

By merging the OSIRIS observations with SSU we determined temperature trends over the 42 years from 1979–2021. The cooling rate for this extended period is about 0.6 K/decade. This is in agreement with the cooling rate in temperatures from SSU merged with MLS and from SSU merged with AMSU-A. The temperature trends in the merged SSU+OSIRIS, SSU+MLS, and SSU+AMSU-A also all agree for 2002–2021. During this period the trends are mainly based on the data records that are merged with SSU, as SSU ceased operations in 2006.

In summary, our results show that the upper stratosphere, from 35-60 km, cooled at a rate of 0.5–1 K/decade between 1979 and 2021. The consistent trends across all observations from OSIRIS, MLS, SABER, SSU, and AMSU-A provide confidence that these cooling trends are accurate. Initial comparisons with reanalysis and model trends highlight the need for further model development in order to accurately represent upper stratospheric temperature changes. The significant stratospheric cooling rate is yet another sign that anthropogenic activities are altering the climate, and it is necessary to model this cooling correctly in order to understand the effects of climate change on the whole atmosphere.

*Code and data availability.* – OSIRIS v7.3 temperature profiles are available at Zawada et al. (2023).

– MLS v5 temperature profiles are available at Schwartz et al. (2020b).

– MLS v5 geopotential heights are available at Schwartz et al. (2020a).

– SABER v2 temperature profiles are available at https://saber.gats-inc.com/data.php

– The SSU v2 temperatures and weighting functions, the AMSU-A v2 temperatures and weighting functions, and the merged v3 SSU+AMSU-A temperatures are all available at https://www.star.nesdis.noaa.gov/smcd/emb/mscat/products.php (NOAA/STAR, 2023).

– SSU+MLS temperatures are available upon request from W. Randel (randel@ucar.edu) (Randel et al., 2016).

– MERRA-2 temperatures are available at Global Modeling and Assimilation Office (GMAO) (2023).

– ERA5 temperatures are available at Hersbach et al. (2023).





- JRA-55 temperatures are available at Japan Meteorological Agency, Japan (2013).

- The WACCM results are available at ftp://odin-osiris.usask.ca/Models. Instructions for downloading the WACCM files are at https://research-groups.usask.ca/osiris/data-products.php#Download

- The LOTUS regression code and documentation are available at (Damadeo et al., 2022).



**Appendix A: Extra Figures**

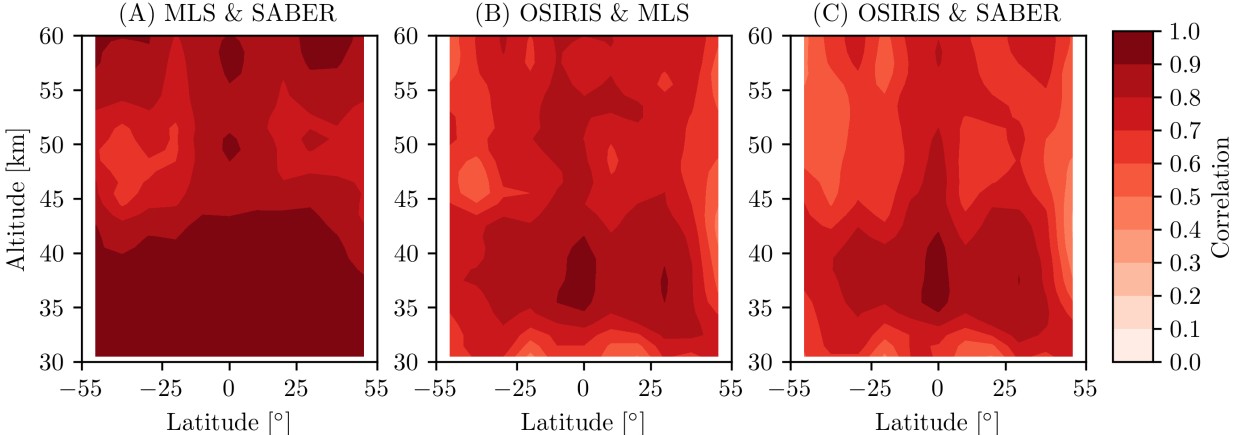

**Figure A1.** Correlation coefficient for deseasonalized monthly mean anomalies during 2005–2021 in 10 degree latitude and 1 km altitude bins. Only months when OSIRIS has observations are considered.





**Figure A2.** Regression coefficients for the solar F10.7 flux and the first two principal components of the QBO. Coefficients are shown for each of MLS, SABER, and OSIRIS temperatures over 2005–2021. Only months when OSIRIS has observations are considered.



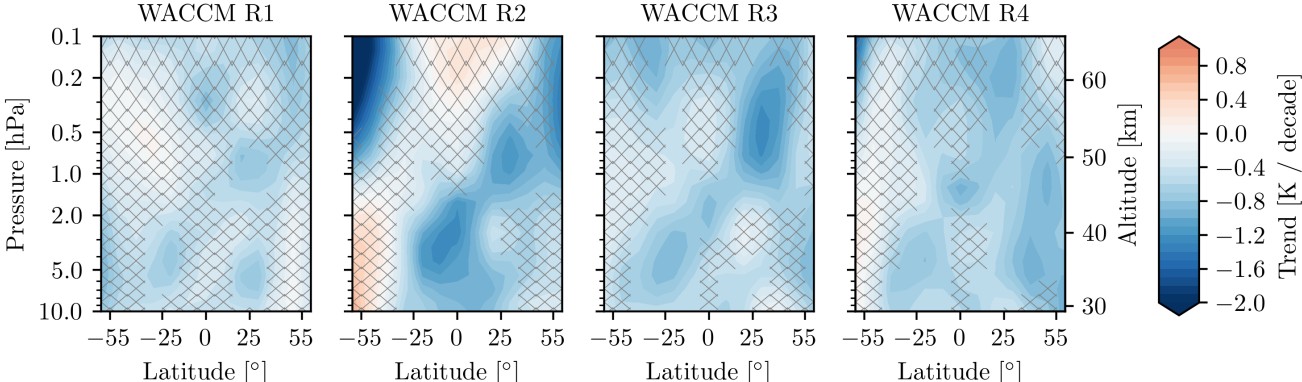

**Figure A3.** Temperature trends for 2005–2018 for 4 WACCM ensemble members. The shaded regions denote the $2\sigma$ uncertainty in the MLR.



*Author contributions.* KD performed the analysis and prepared the manuscript. DZ developed the OSIRIS retrieval. WR provided the WACCM results and the merged SSU+MLS data. ST, AB, DZ, DD, and WR provided input on the method and analysis. ST, AB, and DD supervised the project. SD provided the ERA5 results interpolated to the OSIRIS and MLS profiles. All authors provided significant feedback on the manuscript.

*Competing interests.* We declare that none of the authors have any competing interests.

*Acknowledgements.* This research was supported by the Canadian Space Agency (grant no. 21SUASULSO). The authors thank the Swedish National Space Agency and the Canadian Space Agency for the continued operation and support of Odin-OSIRIS. The National Center for Atmospheric Research is sponsored by the US National Science Foundation. This work was partly supported by the NASA Aura Science Team under Grant 80NSSC20K0928. Work at the Jet Propulsion Laboratory, California Institute of Technology, was carried out under
370 contract with NASA (80NM0018D004).



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
