# Peer review of "Upper Stratospheric Temperature Trends: New Results from OSIRIS"

_EGUsphere, 2024_

## Author Comment (AC1)

**Response to Reviewers: Upper Stratospheric Temperature Trends: New Results from OSIRIS**

**Reviewer 1**

Thank you for the positive comments! Responses to specific points are provided below in blue text.

1) The OSIRIS data are based on a little known technique of deriving the atmospheric density profile and recovering the absolute temperature profile using hydrostatic equilibrium and the ideal gas law. The creation of this dataset is described in detail in Zawada et al (2024). However, a more detailed description of the technique would be useful, outlining in particular the advantages and disadvantages compared with measurement techniques based on the thermal infrared spectrum and microwaves. I recommend also citing other studies based on this technique and in particular the only one that presents a climatology of temperature in the upper stratosphere and mesosphere (Hauchecorne et al., 2019).

We added more detail on the technique to the text and some additional references in this section.  A comprehensive discussion of the advantages and disadvantages of limb scatter instruments versus emission for this particular measurement is perhaps beyond the scope of this paper, especially since most of the advantages are on the instrument building and costing side. But we have tried to emphasize any advantages and disadvantages that are relevant to the trends presented.

2) In the multilinear regression (MLR) model, the quasi-biennial oscillation is represented by two components QBOa and QBOb but without explaining how these components are defined. The MLR model is described in Damadeo et al (2022) but the paper should give sufficient explanation to be consistent.

Further information of the QBOa and QBOb proxies has been added to Section 3:

 "QBO_a(t) and QBO_b(t) are the first two principal components of the monthly mean zonal winds between 300 hPa and 10 hPa measured in Singapore."

3) The way in which the merger with SSU and AMSU-A is carried out should be more detailed. It is not enough to say that it is the same method as Bourassa et al. (2014), although it is important to refer to previous work. Again, the document needs to be

consistent. Where data is available, for example for OSIRIS and SSU for a given month and latitude, do we simply take the average between the two datasets or use a more sophisticated technique?

Thank you for the question. We simply take the average in months/latitudes when OSIRIS and SSU both have observations. More detail has been added to Section 4.2:

"First, the bias between the OSIRIS and SSU temperatures is removed by subtracting the bias from OSIRIS in each latitude bin. The bias is calculated by grouping the observations by month and finding the mean difference for each month when both instruments have observations and then taking the average of these monthly values. Then the datasets are deseasonalized individually to account for differences in their sampling patterns that could affect the seasonal cycle. Finally, the OSIRIS and SSU temperatures are merged by taking the mean in months when both instruments have observations."

4) The temperature trends shown in Figure 8 appear to be more variable with latitude and with greater uncertainty for SSU+OSIRIS than for SSU+AMSU and SSU+MLS. Is this due to less regular sampling with OSIRIS or to greater uncertainties in the OSIRIS data?

The trends in SSU+OSIRIS are likely more variable because of the less regular OSIRIS sampling pattern, as the OSIRIS uncertainty is minimal (~1 K) near 45 km, where the SSU channel 3 weighting function peaks. This is now mentioned in Section 4.2 of the manuscript:

"The SSU+OSIRIS temperature trends are more variable than those from SSU+AMSU-A and SSU+MLS because of the less regular OSIRIS sampling pattern."
* * *
**Reviewer 2**

Thank you for the thoughtful comments on our manuscript! We have edited the text accordingly, and specific comments are addressed below in blue text.

Abstract: You mean probably high-vertical resolution observational data? Otherwise, this statement contradicts with the overview section (SSU, MSU AMSU data are available for a long period of time), and also MLS and SABER profiles are available. Also it is worth to be more specific about the term "long-term

The first sentence has been changed to "Temperature trends in the upper stratosphere, particularly above ~45 km are difficult to quantify due to a deficit of observational data with high vertical resolution in this region that span multiple decades"

Line 39: Please specify the vertical resolution of this dataset

The sentence is now "Randel et al. (2016) also created a merged SSU+MLS data record covering 1979 to the present, however its vertical resolution is limited to that of the three SSU channels."

Line 48: I believe, the abbreviation of OSIRIS is Optical Spectrograph and InfraRed Imaging System

You are correct, the definition of OSIRIS has been fixed.

Line 67: Please provide here a short description (several sentences) of the OSIRIS temperature retrieval algorithm. Please provide also the information about the vertical resolution and estimated uncertainties. This will simplify also reading the next paragraph.

Further details of the OSIRIS temperature retrieval have been added to Section 2.1.1.

Line 108: Please provide here also the information about approximate width of the weighting functions (which defined the vertical resolution of the SSU data)

More information has been added: "The weighting functions of the channels peak near 30, 39, and 45 km, and have vertical resolutions (calculated as the full width at half maximum) of 19, 17, and 15 km, respectively."

Line 169: You apply the regression on deseasonalized anomalies, while you keep the seasonal dependence of QBO. Necessity of including seasonal cycle for QBO is not evident, so please provide more explanation for this or references.

We found that deseasonalizing the data by removing an annual cycling does not account for the coupling between the QBO and the annual cycle. Including the seasonal harmonics for the QBO results in a model that explains more of the variability in the data. We have elaborated on this in the manuscript:

"Seasonal harmonics are nonetheless included for the QBO predictors, QBO_a(t) and QBO_b(t), to account for coupling between the QBO and the seasonal cycle. It was found by Dube et al. (2020) that the MLR does not capture the full QBO signal in the mid-stratosphere when the seasonal harmonics are not included, even if the data has been deseasonalized, because the extratropical QBO signal is modulated by the annual cycle (e.g. Gray et al. 1990, Randel et al. 1999)."

Line 201: Are trends evaluated on 1 km vertical grid? Each altitude independently?

Yes, the following sentence has been added: "Trends are calculated independently at each altitude (every 1 km from 34.5 km to 59.5 km)."

Line 213: Perhaps, the MLS experts, who are co-authors of the paper, could provide more information about possible reasons for the oscillatory pattern in the MLS temperature trends?

It is not clear why the MLS temperature trends have this pattern, but we hope the temperatures will be improved in Version 6 of the MLS retrieval. Version 6 of the MLS data processing is now running, with adjusted filterbank responses that allow us to use more radiances for the temperature retrieval in the uppermost stratosphere and lower mesosphere. We expect version 6 temperature to be better than version 5 temperature in the stratopause/lower-mesosphere region, with better vertical resolution and somewhat less dependance on the temperature prior. It is not clear what impact these changes will have on trends, and analysis of v6 trends will not be possible until reprocessing is closer to completion, likely sometime early in 2025.

Line 214: QBO term has seasonal cycle. Please specify what is shown in FIgure A2 as "QBO".

"The QBO coefficients that are shown are the zeroth order terms, $\beta_{qboA}^{0}$ and $\beta_{qboB}^{(0)}$."

Line 258: I would say simply "similar" Done

Line 283: Is this the bias between monthly zonal mean values? Yes, this has been clarified.

Line 310: The same comment as for the abstract: this sentences needs editions, as several datasets are available.

The sentence now refers specifically to datasets with high vertical resolution.

Figure 6: Are the bias uncertainties very small? If yes, please provide the numbers. If not, please indicate them by errorbars.

Error bars have been added showing the standard deviation of the mean bias.

Figure 7: Please use more distinct colors for SSU+AMSU and SSU+MLS, for example red and green.

Red and green lines together will not pass the journal's colorblindness check. We have instead changed the line colours in the plot to green, orange, and purple, as in Figure 8. Hopefully this is better, it is difficult to choose colours for observations that agree so well!

Data availability: I think, it would be better to provide direct links to the datasets.

The journal policy is that data sets are included in the reference list whenever possible: https://www.atmospheric-chemistry-and-physics.net/policies/data_policy.html

The reference list includes DOIs that function as links to the data.